# In Silico Exploration of the Trypanothione Reductase (TryR) of *L. mexicana*

**DOI:** 10.3390/ijms242216046

**Published:** 2023-11-07

**Authors:** Francisco J. Barrera-Téllez, Fernando D. Prieto-Martínez, Alicia Hernández-Campos, Karina Martínez-Mayorga, Rafael Castillo-Bocanegra

**Affiliations:** 1Departamento de Farmacia, Facultad de Química, Universidad Nacional Autónoma de México, Ciudad de México 04510, Mexico; 2Instituto de Química, Unidad Mérida, Universidad Nacional Autónoma de México, Carretera Mérida-Tetiz, Km. 4.5, Ucú 97357, Mexico; 3Instituto de Investigaciones en Matemáticas Aplicadas y en Sistemas, Unidad Mérida, Universidad Nacional Autónoma de México, Sierra Papacal, Mérida 97302, Mexico

**Keywords:** *L. mexicana*, trypanothione reductase, hotspot, cryptic site, cosolvent molecular dynamics, metadynamics, in silico

## Abstract

Human leishmaniasis is a neglected tropical disease which affects nearly 1.5 million people every year, with Mexico being an important endemic region. One of the major defense mechanisms of these parasites is based in the polyamine metabolic pathway, as it provides the necessary compounds for its survival. Among the enzymes in this route, trypanothione reductase (TryR), an oxidoreductase enzyme, is crucial for the *Leishmania* genus’ survival against oxidative stress. Thus, it poses as an attractive drug target, yet due to the size and features of its catalytic pocket, modeling techniques such as molecular docking focusing on that region is not convenient. Herein, we present a computational study using several structure-based approaches to assess the druggability of TryR from *L. mexicana*, the predominant *Leishmania* species in Mexico, beyond its catalytic site. Using this consensus methodology, three relevant pockets were found, of which the one we call σ-site promises to be the most favorable one. These findings may help the design of new drugs of trypanothione-related diseases.

## 1. Introduction

Leishmaniasis is a neglected tropical disease caused by protozoan parasites, often transmitted by hematophagous sandflies [1,2]. This parasitemia has high levels of morbidity and mortality, with a wide spectrum of clinical complications [3]. Between 700 thousand to 1 million cases of *Leishmania* infections are reported yearly [4]. This figure can increase substantially, considering its cutaneous, mucocutaneous, visceral and kala-azar variants [5]; in fact, 350 million people may be at risk of infection [5,6], with 12 million being actually infected [7].

In this context, drugs have been developed to combat leishmaniasis; however, their indiscriminate use has caused resistance issues [8,9,10]. Moreover, most of these treatments have undesirable side effects, which often turn into toxicity concerns [5,11,12,13]. Nevertheless, these treatments persist as a main course of action, as do sodium stibogluconate/Pentostam^®^ and meglumine antimoniate/Glucantime^®^ (TryR inhibition, decreased parasite reduction capacity [9,14,15,16] and subsequent oxidative stress [17]), amphotericin B (increases parasite cell permeability, causing death by ingress of ions [18,19]), miltefosine/Impavido^®^ (inhibition of phosphatidylcholine [20] and acyl-coenzyme A biosynthesis [21]), paromomycin (respiratory dysfunction, mitochondrial membrane depolarization [22,23,24] and ribosomal damage [25]), and pentamidine (binding to tRNA [26], interference in the metabolism of polyamines [27], inhibition of some transporters [28], and mitochondrial topoisomerase II [29]). Accordingly, target validation efforts are valuable for the discovery of novel and safer treatments anti-leishmaniasis agents [30].

Among the different variants of this disease, the cutaneous one is of particular interest to our research group. Almost 95% of cutaneous leishmaniasis cases occur in the Americas, Mediterranean basin, the Middle East, and Central Asia [4]. Although Mexico is not among the most affected countries by leishmaniasis, it is notably prevalent in its population [31]. The species *L. mexicana* causes up to 90% of the cases reported in the country [31], making this parasite an important object of study.

Several metabolic pathways have been examined as means of parasite inactivation, including, but not limited to: sterol biosynthesis, glycolytic pathway, purine salvage, glycosylphosphatidylinositol (GPI) glycolipid biosynthesis, folate biosynthesis, topoisomerases, hypusine, trypanothione (T(SH)_2_) and the polyamine pathway [32]. The latter two comprise the so-called polyamine–trypanothione metabolism [33] and have drawn the attention of research groups in recent years [34,35,36]. This pathway produces key metabolites that perform crucial physiological functions, such as cell growth and macromolecular synthesis [37], and stabilization of interactions with nucleic acids [38,39]. *L. mexicana* and other parasites start polyamine synthesis from L-arginine, which is converted to ornithine by arginine deiminase (ARG); subsequently, the resulting substrate is transformed to putrescine by ornithine decarboxylase (ODC). Spermidine follows from the conversion of putrescine by spermidine synthase (SPSyn) in presence of S-adenosylmethionine decarboxylase (ADOMETDC). Finally, the latter is converted to T(SH)_2_ by the action of trypanothione synthetase (TryS) [40]. This metabolite is essential for parasite survival, as it is responsible for its response towards oxidative stress from the environment [41]. Therefore, like glutathione for humans, *Leishmania* parasites depend on trypanothione regeneration [42]. It has been shown that *Leishmania* is remarkably sensitive to reactive oxygen species (ROS), in the absence of the T(SH)_2_/TryR system [43]. Here is where TryR comes into play, as the main regulation and regeneration mechanism of this metabolite, and thus is one of the few validated pharmacological targets in *Leishmania* and *Trypanosoma* parasites [44]. Hence, the overall sensibilization to hydrogen peroxide produced by macrophages during infection and the significant differences between host and parasite enzymes, make TryR a promising target for parasite-specific drug development [45,46]. Still, one of the disadvantages of TryR is that the survival rate of the parasite is only affected when the activity of the system is reduced to 90% or more [43,47]. It follows that competitive inhibitors must possess very high binding affinity.

Our research group recently published a pioneering work on this topic, which resulted in the discovery of compound ZINC12151998 (Figure 1), achieving 32% inhibition of the TryR of *L. mexicana* (TryRLmex) at a concentration of 100 µM. Additionally, it showed lower toxicity when compared to amphotericin B and higher activity than Glucantime^®^ [48]. Thus, compound ZINC12151998 is a promising candidate for hit-to-lead optimization; to contribute to the worldwide efforts to identify selective and potent competitive inhibitors of TryR, many of which are reported in the Protein Data Bank [49,50].

Structurally, TryR is a homodimer containing two FAD molecules and two NADPH molecules as cofactors (Figure 2). Each monomer is formed by three different domains: the FAD binding domain (residues 1–160 and 289–360), the NADPH binding domain (residues 161–289), and the interface domain (361–488), which possesses a consistent catalytic site at amino acids Cys52, Cys57 and His461′, respectively [51].

To date, TryRLmex has not been crystallized, making it a virtually unexplored target. Therefore, most reports in the literature include in silico characterizations [52]. However, they often lack comprehensive descriptions or methodologies to aid in drug design. Here, we present a continuation of the previous efforts of our research group [48]. By taking a consensus approach to analyze the topological features of TryR, we aim to identify significant features that can contribute to the discovery of leishmanicidal agents.

## 2. Results and Discussion

### 2.1. Homology Process of TryRLmex and Its Initial Quality Assessment

We built two homology models, for the first one, we used the Swiss Model server and the second one was built with AlphaFold, the latter being one of the best methods to predict the structure of a protein available as of the writing of this work [53]. The Swiss Model template (TRLMS) was built following similar guidelines to our previous work, based on the PDBID code 2JK6, while in AlphaFold (TRLMA), the highest quality model was used, according to the server metrics. To contrast selected templates with the experimental results, the sequences of the constructed homologues were compared with the PDBIDs codes 2JK6 and 2W0H, since their sequence is conserved up to 90% to that of TryRLmex [48]. The QMEAN4 [54] and QMEANDisCo [55] methods, together with the SAVES server, were used to evaluate the 4-four templates.

The 2JK6 and 2W0H showed superior results to TRLMS, but similar ones with respect to TRLMA in all the quality standards used. QMEANDisCO, due to its superiority to QMEAN4 [56] and RMSD, were the main parameters to determine that TRLMA was superior to TRLMS (see Appendix A); yet, the fact that TRLMS showed promise in previous work [48], we determined its quality suffices for further testing.

### 2.2. Analysis of Templates by Molecular Dynamics

To test whether the differences between TryRLmex (TRLMA and TRLMS) and TryRLinf (2W0H and 2JK6) models have a deeper impact, we decided to strengthen the analysis of the models using molecular dynamics and thereby assess the relative stability and overall quality of the models using the classic representative metrics such as RMSD, RMSF and radius of gyration. In addition, the Ramachandran number (𝓡) as implemented in the Backmap library was used [57]. The latter is based on an indexing system that arises from the chirality of amino acids and thus the values for angles ϕ [57]. It has been shown that this proposal for 𝓡 displays smooth relationships towards other structural metrics such as radius of gyration and end-to-end distance [58].

Based on RMSD, RMSF, and radius of gyration of alpha carbons (see Appendix A), no significant structural changes are observed in all models. However, some notable differences were found between crystal structures and homology models; the latter showed lower fluctuations in general. TRLMA would seem more stable than TRLMS based only on RMSD and RMSF values, suggesting a higher quality; however, the radius of gyration trend showed rather abrupt changes on each monomer of TRLMA. Instead, TRLMS showed more consistency and less deviation, making it a good candidate for further modeling.

Ramachandran number plots (see Appendix A), on the other hand, showed that secondary structure remains consistent during simulations. Secondary structure distribution was similar for all models, suggesting a conservation of structural motifs (see Appendix A). This trait was also consistent with the overall displacement and fluctuation of these structural elements during the initial simulation (see Appendix A). These results gave us enough information to choose the structure of the Swiss Model over the AlphaFold.

### 2.3. Hotspot Prediction

Based on the results from molecular dynamics simulations, we chose the TRLMS model and its experimental analogue 2W0H to continue the characterization of this target. Then, we explored the potential of the surface to capture ligands by mapping the presence of putative hotspots. These small regions are crucial to track and identify binding sites [59], since they are the main contributors to the free energy of binding of a molecule to that site [60,61,62]. Their low sensitivity to conformational changes allows them to be identified in almost any structure of protein, including those that do not have a bound ligand [63,64,65,66].

The FTMap server (https://ftmap.bu.edu/login.php (accessed on 24 April 2023)) was used for hotspot mapping [59]. Briefly, FTMap uses 16 organic molecules as probes to examine the macromolecular surface and masks residues belonging to cofactors (FAD and NADPH sites in TryR) (see Appendix A).

In the analysis of our models, most of FTMap probes were concentrated in the targets’ interfaces, with a minority showing preference for the catalytic cavity. The difference between both templates was minimal (see Appendix A). It was observed that FTMap flagged many of the important subpockets in the catalytic cavity (catalytic site, hydrophobic cleft, Z-site, Y-site, γ-glutamic site, and 18-glutamic site) [67,68,69,70,71] and the interface [72,73,74] (see Appendix A) as hotspots.

The ability of these sites to be critical in the reception and accommodation of a ligand is widely known; however, results showed one small pocket that caught our interest. It is a well-defined zone located in the catalytic cavity, comprised of residue 61 (chain A) and residues 396, 398, 399, 462, 463, and 464 (chain B) or vice versa. According to FTMap, it presented van der Waals and hydrogen bond interactions with the probes. Henceforth, we will refer to this area as the sigma site (σ-site) to facilitate its description. This is important because few co-crystallized inhibitors seem to bind to this pocket in close proximity to the His461-Glu466 pair (PDBIDs codes 2YAU, 5EBK, 7NVP, 5SA1, 5SA3, 5S9V, 5S9W and 5S9Z), which is a vital part of the catalytic mechanism [69]. More importantly, it has been observed that in TryR crystals with trypanothione present, the glutathione fragment of the substrate seems to have a notable preference to be placed at this site (PDBIDs codes 1BZL, 1TYP, 2WOW and 4ADW).

Therefore, we sought to gather more data in order to reinforce these preliminary results; thus, we opted to use the Fragment Hotspot Maps server (FHM) (https://fragment-hotspot-maps.ccdc.cam.ac.uk/ (accessed on 6 August 2023)) [75], which is based on fragment hotspot maps created from atomic propensities supported by weighted and remapped donor, acceptor, and hydrophobic probe grids. Furthermore, it offers several advantages over FTMap: it recognizes cofactors and other chemical species, has shorter computing times, and allows visualization of the main area of the pocket and the interactions that drive ligand binding.

FHM highlighted in both templates a similar distribution of hotspots compared to FTMap; in contrast, however, the σ-site showed a poor ranking. Nevertheless, FHM identified two rather attractive zones near the NADPH site. In 2W0H, the important residues were 198, 230, 364, and 374 (first zone) and 377, 378, 423, and 425 (second zone); in TRLMS, only 198, 378, and 425 were relevant (see Appendix A). A fragment-based crystal analysis, performed by Fiorillo et al. on TryR from *T. brucei* (TryRTbru) [76], confirms that the sites flagged by FHM are suitable to receive small ligands (PDBIDs codes 5S9T, 5S9U, 5S9X, and 5SA0). In addition, most of the residues mentioned above are found in the “doorstop pocket” (198, 228, 229, 230, 331, 332, 333, 334, 360, 364, 365, 366, 374, 376, 377, 378, 381, and 385), also reported by Fiorillo et al. [76]. Residues found by FTMap and FHM can be seen in Figure 3 and Figure 4.

### 2.4. Cryptic Sites Prediction

Neglecting protein’s flexibility is an overarching problem in finding putative binding sites, as these depend on the conformations of the target. An example of this came from hotspot analysis of 2W0H, where FTMap showed that, apparently, there were no sufficiently favored areas for hot spots in this target. Even though this is not true, as the configuration of the PDB used was decisive for this result. In such circumstances, hotspots are not enough to define appropriate areas to design new molecules and, therefore, it is necessary to rely on an extension of these: cryptic sites; metastable pockets present in a *holo* structure, but not necessarily in an *apo* structure [77]. These tend to be less hydrophobic and more flexible than traditional binding pockets [78] usually found near hotspots [79]. The latter is of particular interest, since it can provide further contrast between FHM and FTMap results.

For this purpose, the CryptoSite server was used, which is based on the selection of 93 pairs of proteins from the Protein Data Bank in which each unbound structure had a site considered cryptic and each bound structure had a functionally relevant ligand bound to the site, all performed within a machine learning context [77]. Although FTMap has a family of servers where FTFlex [59] has the ability to find cryptic sites, CryptoSite has been found to have slightly higher precision [80].

The distribution of probable cryptic sites found by CryptoSite in 2W0H and TRLMS was very similar (see Appendix A). To our benefit, the σ-site and the zones located in the contours of the NADPH site by FHM showed residues with the potential to form cryptic pockets. It should be noted that these were not only found near hotspots, but within them (Figure 5).This suggests that these areas would not only bind ligands for energetic reasons, but also because of the flexibility of the hotspots and their surroundings, providing a double benefit.

In the light of our findings, these areas are particularly attractive for targeted drug design. As such, our interest shifted to the NADPH site since, as of now, only one crystallized inhibitor contacts the border zone of this pocket (PDBID code 6ER5). In addition, we are dealing with an area located near the cofactor, but significantly different from previous reports [81].

### 2.5. Pocket Prediction by DogSite3

So far, we have discussed relatively unknown sites found in TRLMS that could be use for drug design. Hotspots and putative cryptic sites have proven to be very useful tools. However, even if a protein has either, it is not a guarantee that it can be effectively exploited [82].

Another requirement to consider is geometry, since size and shape of the binding sites are key factors to assess binding affinity [83]. Therefore, DogSite3 was included, as it is based on a GRID method that uses a Difference-of-Gaussian (DoG) filter to detect pockets on the protein surface [84]. This allowed us to see if any of the indicated areas by FTMap, FHM, and CryptoSite are geometrically favored.

DogSite3 revealed that the σ-site and the NADPH site boundaries are favorable zones. Once more, the similarity trend in both templates was preserved (see Appendix A). Likewise, some of the identified areas were discarded, since their distance to essential hotspots showed that the presence of a ligand in them would not be significant (PDBID code 6RB5) [85]. Another important finding obtained by DogSite3 is that more than 50% of the areas in the pockets found by the server are lipophilic surfaces, a trait consistent with the literature [86]. The pockets determined as significant by Dogsite3 can be seen in Figure 6.

For the NADPH site, residues 198, 229, and 230 were not marked as a favorable pocket. In contrast, 360, 361, 362, and 446, among others, were the residues reported by the server in both cases; therefore, this made us question any further consideration of this site.

### 2.6. Cosolvent Molecular Dynamics

Previous methods allowed us to extensively examine the surface of 2W0H and TRLMS, finding broad similarities in both structures; particularly, the zone already described the catalytic cavity and the NADPH site were pointed out. However, these approaches do not account for target’s flexibility [79,83,84,87,88], as they use limited databases [88] and rely heavily on protein orientation [83,89]. Moreover, the servers did not yield any significant differences between TRLMS and 2W0H.

To cope with these limitations, we decided to carry out cosolvent molecular dynamics, as a final test for molecular probes and thereby determine further differences, if any, between 2W0H and TRLMS. As the name suggests, cosolvent molecular dynamics uses a discrete concentration of organic solvents with different chemical properties and with high solubility in water, to avoid aggregation phenomena; this allows the study of putative pockets in the protein [90]. Furthermore, it can be applied to all kinds of protein structures without prior knowledge of potential binding sites [91]. It has been shown that cosolvent molecular dynamics yield efficient mapping of pockets when compared to traditional simulations [92].

Cosolvent molecular dynamics protocols have been maturing over the last decade [93,94,95], so it follows that previous reports act as guidelines more than strict methodological steps. Of the plethora of parameters to be delimited, the first step involves proper choice of probes. For this, we considered the structure of ZINC12151998, the lead compound reported in our previous work [48]. It should be noted that ZINC12151998 was initially proposed from a high throughput docking campaign, comprising 600,000 compounds carried out in the PDB 2JK6 [96]. Thus, as a follow up from our previous work, we intended to gain more information that could aid the hit-to-lead process. Therefore, we opted to extract a representative mixture of linear and aromatic fragments from ZINC12151998. In order to assess their pharmacophoric properties, we aim to characterize putative pockets while also providing guidelines for optimization of the original lead. This resulted in the choice of pyrimidine, imidazole, and *N*-methylacetamide (Figure 7). Pyrimidine is present in the structure of the leader compound and can be used as a bioisostere of the pyridine fragment located in the quinoline system. In turn, imidazole was chosen as a pyrazole isomer and *N*-methylacetamide as a fragment of the linker present in the structure.

Following probe selection, we proceeded to delimit simulation times. We used the protocol from Ghanakota et al. [97] as initial reference, since it covered a large number of targets in its application. An equilibration time of 15 ns was established, while production times of 5, 20, and 40 ns were chosen, with each interval spanning multiple replicas (for further details, see Section 3 Materials and Methods section).

Simulations corroborated much of what was already found by the servers, showing that the σ-site and the surrounding areas of the NADPH site provide the flexibility, geometry, and free energy contribution necessary for the probes to be placed in these and, by extension, a ligand. Additionally, simulations uncovered a third area of interest, located in the upper part of the catalytic cavity and made up mainly of residues 336, 339, 340, and 343 (chain A), residues 458, 459, 461, 466, 470, and 472 (chain B), and vice versa. This cavity was partially suggested by FTMap and CryptoSite servers. Henceforth, we shall refer to this as the lambda site (λ-site), which is depicted in Figure 8. Currently, there is no rationally designed ligands reported in the literature that can to bind to the λ-site. Still, PDBID code 1BZL shows that the glutathione fragment of trypanothione has a certain tendency to settle there; thus, on par with the σ-site, it could provide an opportunity to obstruct access into the catalytic groove.

Despite the similarity of results between production times, we set ourselves the task of examining which of these proved optimal to achieve robust analysis and mapped pockets in the shortest simulation time and with the minimum number of replicates. When comparing the surfaces of the pockets found and production times, we observed that there was a great similarity of results between 20 and 40 ns (Figure 9), both in the area of the representative pockets (Figure 9C,D) and in their score. 

On the other hand, 5 ns had significant differences compared to the latter (Figure 9A), despite the fact that more replicates were performed. A broader comparison can be found in the Appendix A. Since the results between 20 and 40 ns did not differ greatly (we call this Experiment A), we opted to use a 20 ns triplicate to compare 2W0H and TRLMS in a second cosolvent dynamics experiment and thereby gather more information (Experiment B and C).

The results showed that the affinity of both templates for selected probes is very similar, yet the distribution of these on the targets’ surface suggests that TRLMS has a greater facility to place probes in the vicinity of the NADPH site and the σ-site than 2W0H. In addition, the latter exhibits more distant locations from critical areas of the target as relevant than TRLMS, which suggests that 2W0H may retain some ligands in these areas before they reach the essential sites related to the inhibition of the target. The residues of each cavity, as well as their location, can be consulted in the Appendix A.

The relevance of the previous results could be observed by comparing the target residues that interacted with the ZINC12151998 in a classical molecular dynamics simulation with a 50 ns trajectory carried out in our previous work [48]. Where residues 110, 466, and 467 interacted with the linker, residues 355 and 459 came into contact with the quinoline moiety; and residue 461 interacted with both the pyrazole and pyrimidine nuclei. All of these belong to important areas of the catalytic cavity, such as the hydrophobic cleft (110), the catalytic site (461), the γ-glutamic site (461, 466, and 467), and the λ-site (459, 461, and 466). Although none of these are found in the σ-site, the fact that the chosen probes interacted in key places in the main cavity allows us to deduce that modifications in the ZINC12151998 fragments should be well-received by the target for binding. The chosen simulation time and the size of ZINC1215998 could have posed an obstacle for more fragments of it to interact with the λ-site.

### 2.7. Metadynamics

Cosolvents molecular dynamics simulations allowed us to determine that the σ-site, the λ-site, and the contours of the NADPH site were the best candidates for rational design of drugs, since they have most of the required features by a cavity to capture and retain a ligand in its vicinity. Due to its proximity to the His461-Glu466 pair, we chose the σ-site among them to carry out a final test: metadynamics. Briefly, metadynamics allows the recovery of free energy values and characterization of folding events using an enhanced potential along a given choice of reaction coordinates, also known as collective variables (CVs) [98].

For this study, we chose the well-tempered variation of the method to avoid the system transients on high-energy regions. For our analysis, six replicas of 50 ns were obtained.

Naturally, this production time is insufficient to yield a reliable estimate on either free energy or the overall shape of the potential energy surface. However, we did not aim want to estimate them precisely, but rather gain intuition into the general flexibility of the region and exploit it to reinforce the feasibility of using this region as a suitable ligand design. The first system to which metadynamics were applied was TRLMS, which showed that the σ-site presents some metastability (Figure 10).

As observed, there is a clear trend for a local minimum, with average values of ~2.5 Å and ~6.5 Å for RMSD and radius of gyration, respectively. Only one replica showed the presence of additional minima, yet such behavior was not observed in 2W0H (see Appendix A). This suggested that the bias factor was high, leading to low resolution of this zone. To account for these, four additional metadynamics runs were performed using different values for bias (3.0 and 8.0). Free energy surfaces of these are shown in the Appendix A.

For comparison, Figure 11 shows surface representations of representative conformations obtained around minima values. It can be seen that there is high flexibility in this region of the interface.

Considering these results, we made a comparison to assess whether metadynamics indeed revealed something that conventional molecular dynamics simulations would not have shown. Based on radius of gyration values, the site shows metastability with small but consistent fluctuations, a behavior also observed in TRLMA (Appendix A). In the light of this, we compared radius of gyration trends from classical and enhanced simulations using histograms (Appendix A). A well-defined minimum around 7 Å is consistent; the slight presence of other local minima is also suggested, but due to the nature of metadynamics, these are filled quickly, and thus there are higher frequency counts. This suggests two things: a high structural integrity of the σ-site and, perhaps more intriguingly, that the conformational evolution of the zone is more complex than initially thought. Of course, further characterization is required, for instance the kinetic transition analysis using Markov state models [99] in TRLMS and 2W0H. Such study is beyond the current scope of this work.

### 2.8. Relevance of the Pockets Found

With the aid of computational tools, highly relevant pockets were found in TryRLmex. However, this by itself is not enough, as it is our intention to explore their significance for drug design, even if preliminary. To do this, we considered TryR from other parasites such as *L. infantum* (PDBID code 2JK6), *T. cruzi* (PDBID code 1BZL), and *T. brucei* (PDBID code 2WBA). Hence, we compared the residues from the σ-site, λ-site, NADPH site, and doorstop pocket, including their contours, to examine the degree of conservation of these pockets; already known areas were also included.

We found that, in general, residues from already known pockets are largely conserved (see Appendix A). Notably, the doorstop pocket, which we consider more significant due to its experimental support [76], showed important variations (198, 332, 364, 377, 378, 381, and 385). The catalytic cavity (102, 105, 112, 347, 356, 402, 455, and 456) and, finally, the NADPH site (166, 167, 288, 361, and 423) followed this. In all cases, the most notable variations were observed in *T. cruzi* and *T. brucei*.

In sharp contrast, the σ-site is also completely conserved among the trypanosomatids mentioned above; thus, it remains a strong candidate to initiate a rational drug design against TryR. In addition, the experimental evidence in the RCSB database shows several crystallized inhibitors that bind to this pocket. Of note, their effectiveness seems not to be conditioned only to binding to the σ-site, but to an interplay with additional sub-pockets in the catalytic cavity (PDBIDs 2YAU, 5EBK, 7NVP). This suggests that the effectiveness of the σ-site as a binding site will also depend on the haptoforic regions of TryR. In this case, almost all of the previously described pockets in the catalytic cavity (catalytic site, hydrophobic cleft, γ-glutamic site, Z-site, etc.) are conserved in the trypanosomatids already mentioned. Thus, the premise of broad-spectrum drug gains higher relevance.

Finally, it is necessary to consider another critical point: the potential risk of collateral inhibition of glutathione reductase (GR), the enzyme analogous to TryR in humans. Although it has been reported that the catalytic cavity of GR is highly conserved with respect to that of TryR [42], there are opposite features in both targets: first, the difference in the orientation of the helices in the FAD domains of TryR and GR make that the catalytic cavity of the first is larger than that of GR [100,101,102] and, secondly, due to the differences between sequences (see Appendix A), TryR presents a more hydrophobic and negatively charged catalytic cavity [100,103,104] than GR, the which is more hydrophilic and positively charged [105]. All these differences lead to a different distribution of charge, size, and structural orientation with respect to TryR. Several research groups reported that these differences have been shown to be crucial when comparing the inhibitory activity of various compounds on TryR and GR, finding that the vast majority prefer the first target without significantly inhibiting GR [74,85,106,107,108,109,110,111,112].

Regarding the σ-site in TRLMS and GR, it is conserved in six out of seven residues, with the only difference being the substitution of Leu399 in TRLMS for Met406 in GR. This by itself does not provide us with much information about the potential behavior of the σ-site in both targets, so we conducted final comparison between proteins using metadynamics and molecular docking.

Using a similar protocol to that of TRLMS, metadynamics showed notable differences between the dynamic behavior of the σ-site (Appendix A). Particularly, higher bias factors showed that the subtle minima present is TRLMS, and it is more noticeable in GR. This, combined with previously mentioned features, suggests that ligands may indeed show different affinities for this sub-pocket. As a proof-of-concept, we examined the behavior of this pocket in both targets in the presence of representative ligands. To do this, ZINC12151998 and compound **71** were selected (Figure 12); the latter being a molecular probe reported by Fiorillo et al. [76] that showed affinity for the σ-site of TryRTbru and bears structural similarities with ZINC12151998. Certainly, molecular docking could not provide a solid criterion; thus, we carried out well-tempered metadynamics to assess the putative affinity and overall stability of the putative binding mode of both compounds in the σ-site.

Both ligands were docked in TRLMS and GR with the help of the LeDock [113,114,115] and PLANTS [116,117,118] programs, ensuring that the grid box occupied most of the catalytic cavity, including the σ-site. For ZINC12151998, the choice of binding mode was based on scoring and the interaction profile with the σ-site, whereas for compound **71**, we relied on the pose found in PDBID code 5S9W (see Appendix A).

The docking calculations showed interesting results. A general view can be seen in Figure 13. 

The docking of compound **71** in TRLMS obtained very similar poses in both programs, observing interactions with Phe396 (π-π) and Leu399 (hydrogen bond) through the phenyl and amide residues of the ligand; it also highlights that the orientation with respect to the native pose of **71** is inverted, but the type of interactions is similar (Figure 13A; see also Appendix A). Docking in the GR showed important changes in the calculated poses, obtaining conformations that introduce the furan fragment in the σ-site (Figure 13B). In this case, no relevant interactions were presented, except the approach of the ligand to residues Lys67, Met406, and Ser470. The docking of ZINC12151998 in TRLMS showed in both programs poses with a hydrogen bond at residue Leu399 through the pyrimidine fragment (PLANTS) or pyrazole (LeDock); the rest of the compound structure is distributed in the σ-site, and only one π-cation interaction found by PLANTS at residue Lys61 with one of the aromatic fragments of the structure stands out (Figure 13C). The conformations of Lys61/Lys67 in the PDBIDs suggest that this residue serves as a “gatekeeper” that regulates entry to the σ-site. As with **71**, docking of ZINC12151998 in GR with PLANTS suggests a partial introduction of the ligand to the σ-site supported by a hydrogen bond with residue Ser470 while LeDock placed the ligand at the boundary of the cavity through a hydrogen bond with Thr404 and Met406 (Figure 13D). This last residue, which differs in TRLMS, seems to present a conformation that, like Lys61/Lys67, regulates access to the σ-site. The GR substitution of Leu399 for Met406 could increase the overall flexibility of the pocket by inducing some selectivity in the binding of a ligand [119]. Once the poses of each ligand were chosen, we examined the metastability of the σ-site in both cases.

Metadynamics with both ligands showed notable differences in the obtained free energy surfaces. Of note, metadynamics with ligands gives rough estimates of affinity, but can yield valuable information on binding mode stability [120]. In our case, we selected high bias parameters in an attempt to uncover the interplay between sub-pockets, if any. Such was the case for compound **71**, as metadynamics showed that there is significant affinity for the σ-site and its vicinity (Figure 14), with the binding mode from PLANTS showing four basins that can be considered local minima around this initial arrangement.

LeDock’s pose, on the other hand (Figure 15), showed less stability, with a tendency for minima around 1 nm from the σ-site, which comprises a hydrophobic cleft where Leu399 is found. This, with the addition of the observed hydrogen bonding, suggests an interplay between the gatekeeper residue and Leu399 for ligand orientation.

In contrast, simulations of **71** in GR showed significant differences. For the case of PLANTS, only two basins were identified, with similar affinities. In this case, it was observed that **71** indeed left the overall vicinity of the σ-site, accounted for as “unbinding” for our purposes (Figure 16).

Similarly, LeDock’s pose metastability was found to be around 10–14 Å from the starting pose; in this case, it suggests that for compound **71**, affinity is away from the gatekeeper Lys67, leading to another unbinding event (Figure 17). Overall, this serves as evidence that GR showed a different recognition of this fragment, considering the preservation of the σ-site; this further supports the interplay between pockets in TryR and the different features of these when compared to GR.

For ZINC12151998, the metadynamics in TRLMS showed an unbinding event for PLANTS’ pose (Figure 18).

This could be attributed to the size and overall hindrance introduced by the seven-membered ring. Notably, in this case π-cation interaction with the gatekeeper was observed, giving further hints on its role for binding stability. In contrast, the metadynamics of the LeDock pose did not show this interaction, and had lower affinity estimates, plus also showed an unbinding event (Figure 19).

In contrast, the ligand metadynamics in GR from PLANTS’ pose showed even lower affinities and no indication of metastability (Figure 20 and Figure 21). 

Another significant difference is that the GR 2D diagrams show a better fit of the ligand within the homodimer interface, suggesting a volume difference. Yet, this serves to further underline the interplay of haptoforic features, resulting in no proper orientation of the ligand within the cavity. Considering LeDock’s pose, an increase in affinity and some metastability was observed; however, this arrangement involves mostly polar residues and regions in the vicinity of the σ-site, suggesting vague contacts due to the size of ZINC12151998.

## 3. Materials and Methods

### 3.1. Homology Modeling and Quality Evaluation of TryRLMex

The TryR sequence was obtained from the UniProt server (https://www.uniprot.org/ (accessed on 10 August 2019)) [99] entry E9AKE1, gene name LMXM_05_0350, organism *L. mexicana* (strain MHOM/GT/2001/U1103). This sequence was submitted to the Swiss Model server (https://swissmodel.expasy.org/ (accessed on 12 August 2019)) [121] and, from the models obtained, the one based on the PDBID code 2JK6 was chosen, as it showed significant quality among the output models. In the same way, the mentioned sequence was uploaded to the AlphaFold structural prediction tool [122,123] included in ChimeraX (v.1.6.1) [124,125,126], and only the highest quality model according to AlphaFold metrics was chosen. Additionally, PDBIDs codes 2JK6 and 2W0H, belonging to the TryR of *Leishmania infantum* (*L. infantum*), were also downloaded for comparative purposes [50]. The Protein Preparation Wizard (PrepWizard) in Maestro (v.2021-4) [127] was used to prepare the enzymes in this study. The systems were protonated at physiological pH value using PROPKA [128], and the hydrogen bonding network was optimized with the ProtAssign function [129]. Quality was assessed with the SAVES server (https://saves.mbi.ucla.edu/ (accessed on 14 august 2019)) [130,131,132,133,134] and the Qualitative Model Energy Analysis) (QMEAN) metrics of the Swiss Model [135].

### 3.2. Dynamic Analysis of Templates

Molecular dynamics simulations and its visualization were performed with Desmond (v.2021-4) [54,127], beginning with a simulation of 200 ns for TryR homologous models and PDBIDs 2JK6 and 2W0H, performed to verify the structural integrity of the models. Proteins and cofactors were parametrized with the OPLS_2005 forcefield [129]. The simulation cell in all cases comprised an orthorhombic box buffering the enzyme by 12 Å, the water model was TIP3P [136], and sodium chloride was added to obtain a concentration of 0.15 M. Minimization was carried out in three steps, using Brownian dynamics at 10 K. The first stage involved a 250 ps simulation with restraints on solvent-heavy atoms with a force constant of 100 kcal/mol. This was followed by 250 ps using restraints on the enzyme backbone with a force constant of 50 kcal/mol. Finally, a minimization run without restraints was carried out for 1 ns. Each system was then equilibrated using the default relaxation protocol in Desmond. Integration employed the RESPA algorithm [137], with hydrogen mass repartition used to increase timestep to 4 fs for bonded and near interactions, whereas far interactions were integrated every 8 fs. Production runs were performed under NPT ensemble at 300 K using the Nosé–Hoover chain thermostat [138] and MTTK barostat [139].

These simulations were then analyzed using Backmap (v.0.2.6) [57] to assess the behavior of the models and overall distribution of secondary structure. Additional analyses were performed using MDTraj (v.1.97) [140].

### 3.3. Preliminary Search for Hotspots

Mapping and searching for hotspots was performed using the FTMap server (https://ftmap.bu.edu/login.php (accessed on 24 April 2023)) [59]. For the correct analysis of the targets, the masking option was used, focusing on the residues belonging to the FAD and NADPH sites indicated in the PDB file built with Maestro, according to the server’s instructions. Visualizations of the results were made with Open-Source PyMol (v.2.5.5) [141].

Additionally, the Fragment Hotspot Maps server (FHM) (https://fragment-hotspot-maps.ccdc.cam.ac.uk/ (accessed on 6 August 2023)) [75] was used to reinforce the information found by FTMap. PDB files previously prepared in Maestro were entered into the server and only residues with a cutoff of 17 (strong hotspot) and 14 (weak hotspot) were considered.

### 3.4. Search for Cryptic Sites

The search for cryptic sites was performed with the CryptoSite server (https://modbase.compbio.ucsf.edu/cryptosite/ (accessed on 30 April 2023)) [77]. PDB files previously prepared in Maestro were entered into the portal and, once the results were obtained, only residues with a CryptoScore of 10 or higher were taken into account, in accordance with the recommendations of the portal. The visualizations of the results were made with Open-Source PyMol [141].

### 3.5. Pocket Search

The prediction of pockets and their relevance was carried out with the DoGSite3 server (https://proteins.plus/#dogsite (accessed on 3 May 2023)) [84]. PDB files previously prepared in Maestro were entered into the portal. The server was adjusted to analyze only pockets. Visualizations of the results were made with UCSF ChimeraX [126].

### 3.6. Cosolvent Molecular Dynamics

The search for pockets and hot spots using molecular dynamics was carried out with Desmond [127]. For this, a modified version of the Ghanakota et al. protocol was used [97]; briefly, the protocol involves the creation of replica systems with a volume/volume concentration of 95/5% (water/cosolvent). Systems were constructed using the MxMD utility available in Maestro [142]. It uses preequilibrated boxes of the selected cosolvent probe, which is then incorporated in a cubic box of water buffering the whole system by 15 Å. These systems were then submitted to a relaxation protocol and 15 ns of equilibration time. To improve sampling and overall exploration, different production times were chosen for TRLMS: 5, 20, and 40 ns. A total of 60 simulations were performed, accounting for more than 650 ns of total simulation time (Experiment A). Three main probes were chosen: imidazole, pyrimidine, and *N*-methylacetamide; these were selected by fragment analysis of ZINC12151998 [48]. Hotspots were then evaluated by the following score:(1)$$Score=\sum_{p}^{probes}\sum_{s}^{spots}\sum_{g}^{grids}Z\left(p,\;s,\;g\right)$$
where *Z_xyz_* is the occupancy of functional groups of the probes, converted to *Z*-*scores* using the following equation:(2)$$Z_{xyz}=\frac{X_{xyz}-\langle X\rangle }{\sigma }$$
where *X_xyz_* comes from the raw bin count for each grid and 〈*X*〉 is the mean value for all the bins and *σ* is their standard deviation [143].

Additionally, the same protocol was applied to 2W0H and TRLMS, with a production time of 20 ns (Experiment B and C). A total of 3 simulations per template were performed, accumulating 60 ns per system, for comparison purposes. Visualizations of the results were made with UCSF ChimeraX [126].

### 3.7. Metadynamics Simulations

To test the behavior of the σ-site, we conducted enhanced sampling using the well-tempered metadynamics algorithm [144]. Briefly, this method enhances sampling with the use of collective variables, which are biased using a theoretical construct similar to adaptive umbrella sampling. In the case of metadynamics the histogram of the CV is approximated using its logarithm with the aid of a simple differentiable function; i.e., a Gaussian potential of a given height and width (*σ* and *w*, respectively);
(3)$$T\;\log\left[H_{t}\left(s\right)\right]\sim \;w\;\exp\;\left[\frac{{\left(s_{t}-s\right)}^{2}}{2\sigma^{2}}\right]$$

To date, there is no systematic nor explicit way to determine what may constitute a pertinent choice for CVs; reasonable choices should still be robust enough to distinguish reactants, transition state, and products [145,146]. For this case, collective variables were the RMSD and radius of gyration of the center of mass composed by residues 58 and 61 from chain A and residues 396, 398, 399, 462, 463, and 464 from chain B. This interface region has been proposed as a putative site of pharmaceutical interest [73]. The parameters for the Gaussian functions were a height of 0.03 kcal/mol and a width of 0.15 Å for both variables. The rate at which Gaussians were deposited was 1 ps, the bias factor for the well-tempered algorithm was set to five. Briefly, the bias factor or Δ*T* is the heart of the well-tempered algorithm, as it establishes the rate at which the Gaussian height decreases. It follows that for ΔT→0, conventional MD is recovered, and the limit ΔT→∞ corresponds to standard metadynamics implementation, thus the fine-tunning of the bias parameter restricts the exploration of high energy states, which may result in their being irrelevant [98].

Simulations accounted for 50 nanoseconds; additional settings were kept from the initial simulations. Six replicas were obtained under these conditions for the homology model obtained from the Swiss Model and 2W0H for comparison. Free energy surfaces were analyzed using Maestro.

### 3.8. Docking

Molecular docking of ZINC12151998 and compound **71** was carried out with the PLANTS and LeDock programs, recognized for their precision and accuracy [147,148].

PLANTS uses the well-known ChemPLP scoring function, which comes from:(4)$$f_{PLANTS}=f_{plp}+f_{clash}+f_{tors}+C_{site}$$

The scoring function of the piecewise linear potential function (*f_plp_*) is used to model steric complementarity of the protein and the ligand. To avoid internal ligand clashes, an empirical heavy-atom potential (*f_clash_*) is evaluated for each ligand pose. The torsional potential (*f_tors_*) is calculated for all rotatable bonds available in the ligand molecule except for terminal hydrogen bond donor groups. Finally, as it is outside the binding site definition, which is defined by a sphere in PLANTS, no interaction potentials are calculated, and a mechanism is needed to guide the search algorithm toward the interesting regions located at the predefined binding site. A quadratic potential is calculated for the reference point of the ligand (origin of the ligand’s coordinate system) if this point lies outside the sphere, that is, the distance r of the reference point to the center of the sphere is greater than the binding site radius (*C_site_*) [116]. PLANTS was used with its default parameters for exhaustiveness.

On the other hand, LeDock uses the following equation as a scoring function:(5)$$\begin{array}{c} \Delta G_{bind}=\alpha \sum_{i\in lig}(E_{i}^{vdw}+E_{i}^{hb})\Theta \left(E^{CO}-E_{i}^{vdw}-E_{i}^{hb}\right)+\\ \beta \left(r\right)\sum_{i\in lig}\sum_{j\in pro}\frac{q_{i}q_{j}}{r_{ij}}+\gamma E_{lig}^{strain} \end{array}$$

The first term is the sum of *vdw* interaction *E^vdw^* and hydrogen bonding energy *E^hb^*, where *Θ* is the Heaviside step function and *E^CO^* is the cutoff energy to allow for soft docking. The second term is the electrostatic interaction energy with a distance function *β*(*r*) accounting for both electrostatic screening and desolvation effects, where *q* is the partial atomic charge and *r* is the distance between a pair of atoms. The last term is the ligand conformational strain upon binding, and is contributed by the intra-molecular clash and/or torsion strain. Coefficients *α*, *β*, and *γ* were empirically determined by referring to the previous scoring function [113].

Proteins were prepared using LePro [114] and, for docking, the parameters that were established were an RMSD between poses of 1.0 Å and a grid box with measurements in *x*, *y* and *z* of 25 Å centered on the catalytic cavity of both targets, in such a way that all of it was part of search space.

### 3.9. Ligand Metadynamics

Just like the initial metadynamics simulations, Desmond was used with similar parameters and settings. Selected binding modes from both docking programs were used as starting point for these simulations. In this case, collective variables were defined by the distance between the center of mass from the σ-site and the ligands; the second reaction coordinate was ligand’s RMSD; for both CVs, the width was set to 0.1 Å and gaussians were deposited every picosecond. The bias factor was set to 8.0. Two replicas of 20 ns were carried out for each ligand and program yielding four simulations for comparison; additional FES diagrams can be found in the Appendix A.

## 4. Conclusions

In the present work, the analysis and comparison of a homologue of the TryR from *L. mexicana* was carried out. The study included the search for hotspots, cryptic sites, and overall druggable pockets in static models, later reinforced with molecular dynamics simulations and enhanced sampling. To the best of our knowledge, this is the first study of its kind on this target. Our study sought to find pockets that satisfied the following criteria: binding free energy contribution, flexibility, and a favorable geometry. Despite not finding significant differences between the homologue and 2W0H, we found three putative pockets, two of which were found in the vicinity of the catalytic cavity and one in the vicinity of the NADPH site. Certainly, we consider that, due to its proximity to the His461-Glu466 pair, the pocket denominated as the σ-site was the most appropriate to perform additional analyses. Preliminary metadynamics studies showed us that this zone presents acceptable metastability and flexibility, desirable characteristics in a pocket that is intended to place a ligand, for which reason we consider this zone critical in the targeted design of drugs. Furthermore, the fact that it remains highly conserved in the trypanothione reductase of other parasites and the broad differences between TryR and GR make this site an extremely attractive area. Finally, enhanced sampling around this pocket suggested a notable cooperativity of previously identified sub-pockets to allow binding around the interface region. Moreover, the role of Lys61 has been highlighted as a putative gatekeeper and haptoforic interaction that allows ligand orientation and overall affinity for the aforementioned subpockets and σ-site.

In summary, this information can serve as a starting point for drug design efforts, since now it would not be necessary to consider the entire catalytic cavity to design an inhibitor, which would facilitate its design.

## Figures and Tables

**Figure 1 ijms-24-16046-f001:**
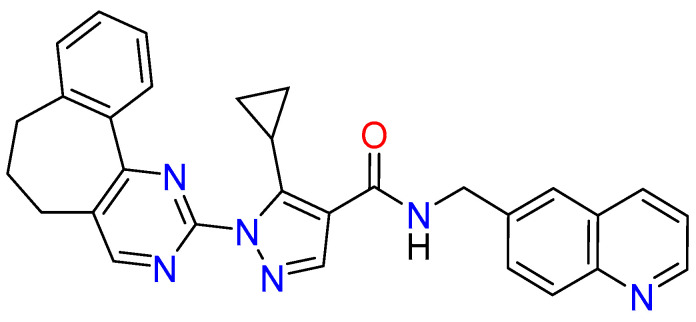
Chemical structure of ZINC12151998.

**Figure 2 ijms-24-16046-f002:**
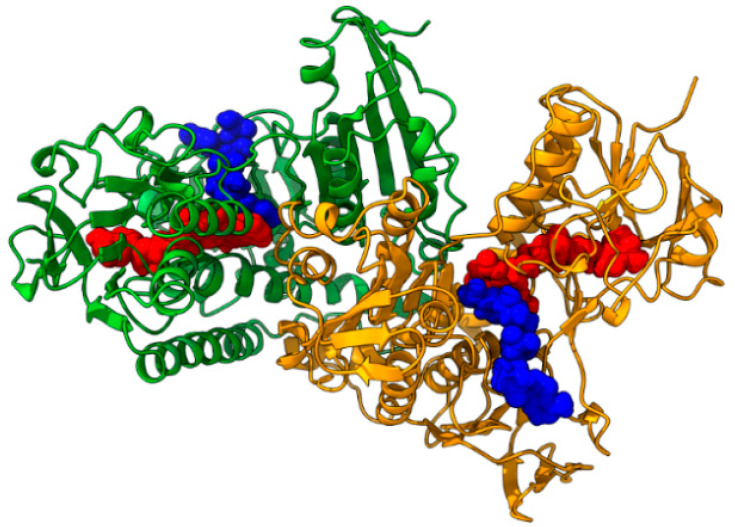
Structure of the TryR of *L. infantum* (TryRLinf), PDBID code 2W0H. Chain A = green, chain B = orange, FAD = red spheres, NADPH = blue spheres.

**Figure 3 ijms-24-16046-f003:**
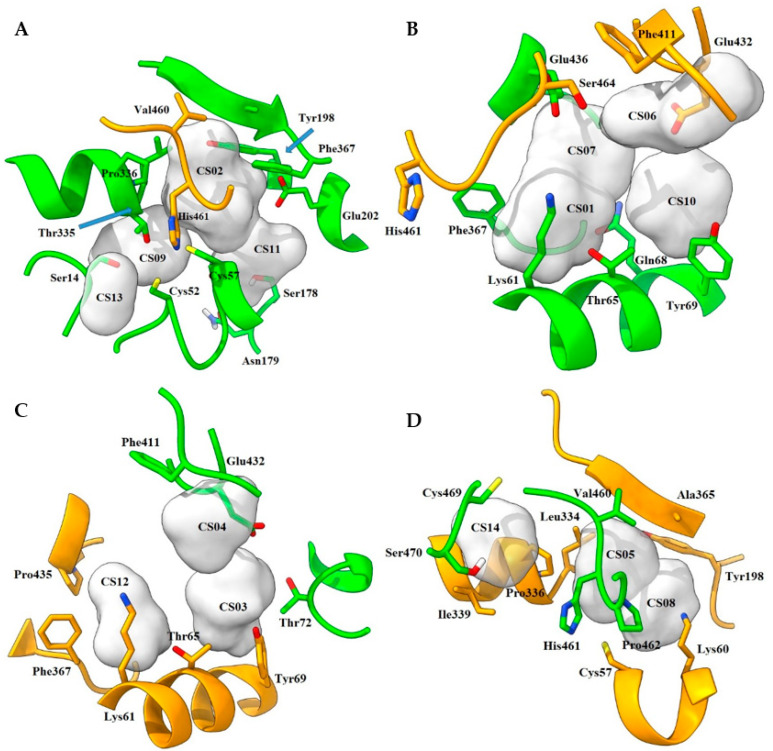
Distribution of hotspots in TRLMS found with FTMap server. Consensus sites (CS tag) are shown as light gray surfaces (see Appendix A). Only the most important residues are shown. (**A**,**D**) catalytic cavity; (**B**,**C**) interface. Chain A = green, chain B = orange. Atom color: N (blue), O (red), S (yellow).

**Figure 4 ijms-24-16046-f004:**
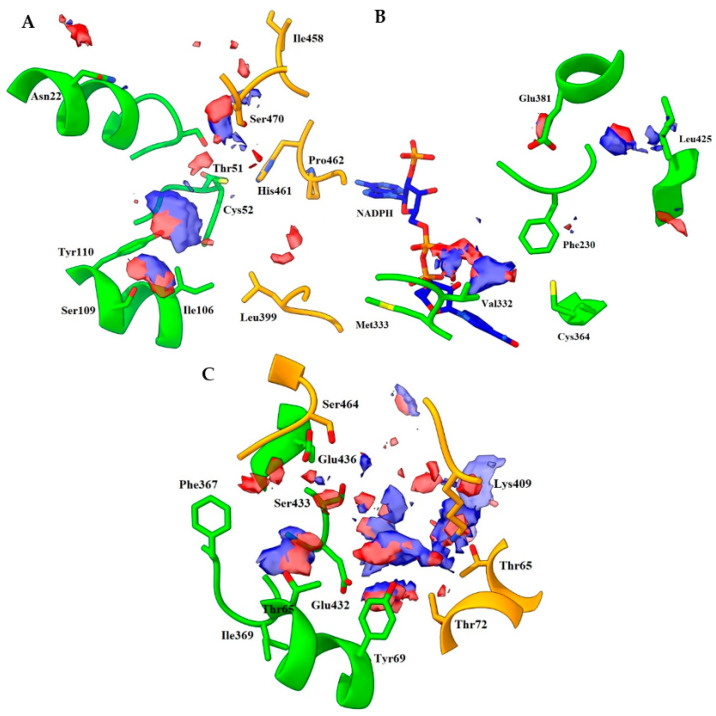
Distribution of hotspots in TRLMS found with FHM server. FHM shows hotspots as surfaces of an electron donor (blue) or acceptor (red) character; the cofactor NADPH (blue and red sticks) can be observed. Only the most prominent residues are shown for clarity. (**A**) catalytic cavity; (**B**) NADPH site; (**C**) interface. Chain A = green, chain B = orange. Atom color: N (blue), O (red), S (yellow).

**Figure 5 ijms-24-16046-f005:**
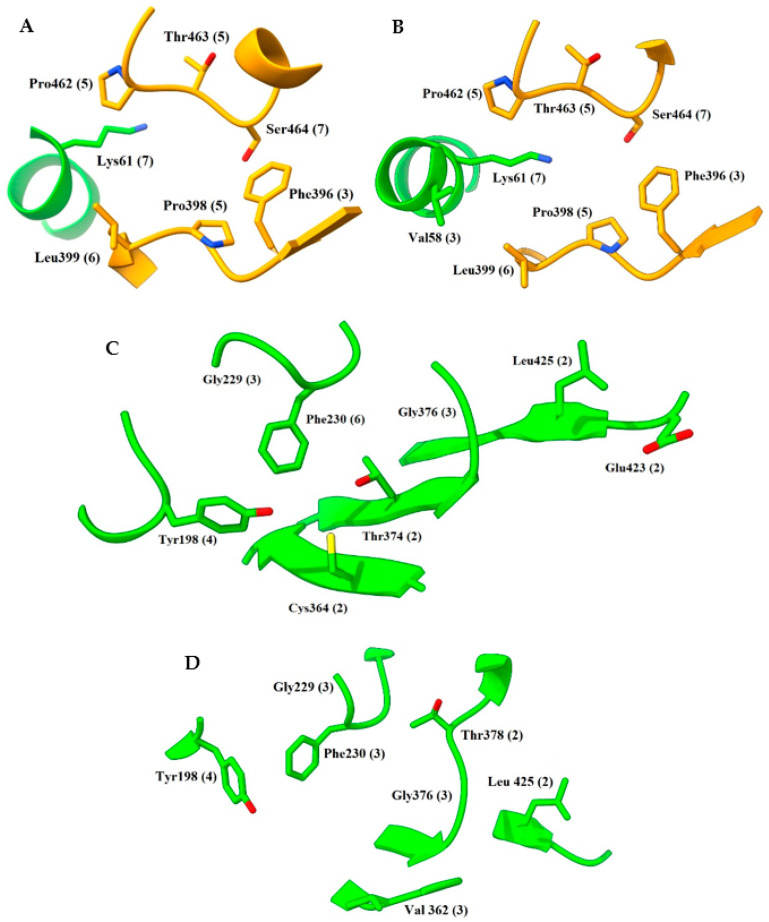
Important residues found with FTMap, FHM, and CryptoSite. (**A**,**B**) σ-site of 2W0H and TRLMS; (**C**,**D**) NADPH site of 2W0H and TRLMS. Residues found either by a single server or by consensus have different tags between parentheses: FTMap (1), FHM (2), CryptoSite (3), FTMap–FHM (4), FTMap–CryptoSite (5), FHM–CryptoSite (6), and FTMap–FHM–CryptoSite (7). Chain A = green, chain B = orange. Atom color: N (blue), O (red), S (yellow).

**Figure 6 ijms-24-16046-f006:**
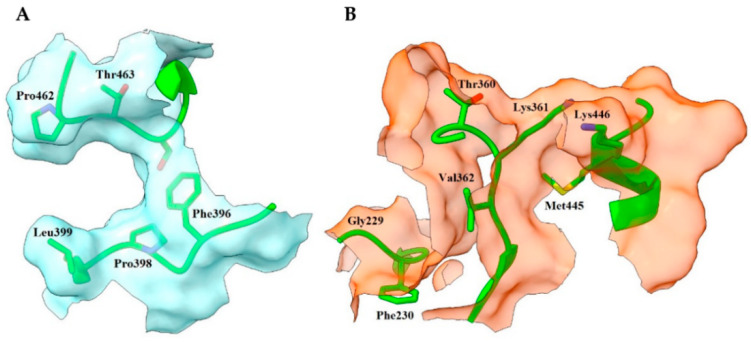
Key residues found by DogSite3 on TRLMS. (**A**) σ-site; (**B**) NAPDH site. Note that the entrance of the cavity in (**B**) is on the side of the catalytic site and not the NADPH site. Chain A = green. Atom color: N (blue), O (red), S (yellow).

**Figure 7 ijms-24-16046-f007:**
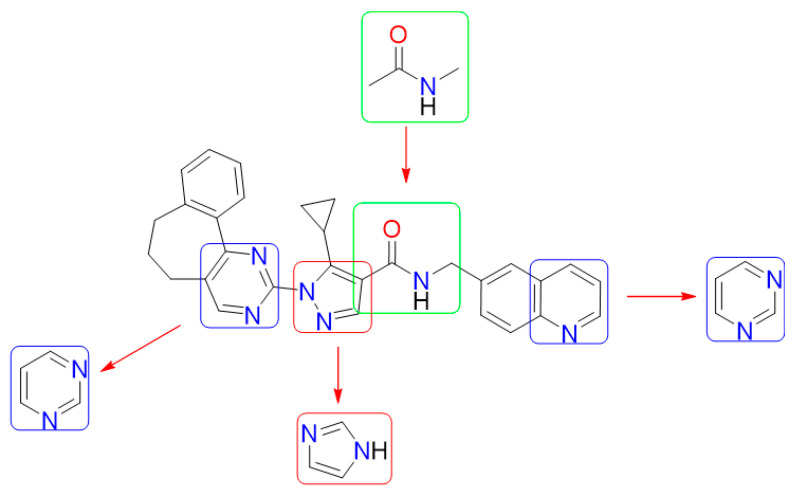
Fragment decomposition of ZINC12151998 for probe selection. Blue, red, and green boxes indicate fragments the same or similar to pyrimidine, pyrazole, and *N*-methylacetamide.

**Figure 8 ijms-24-16046-f008:**
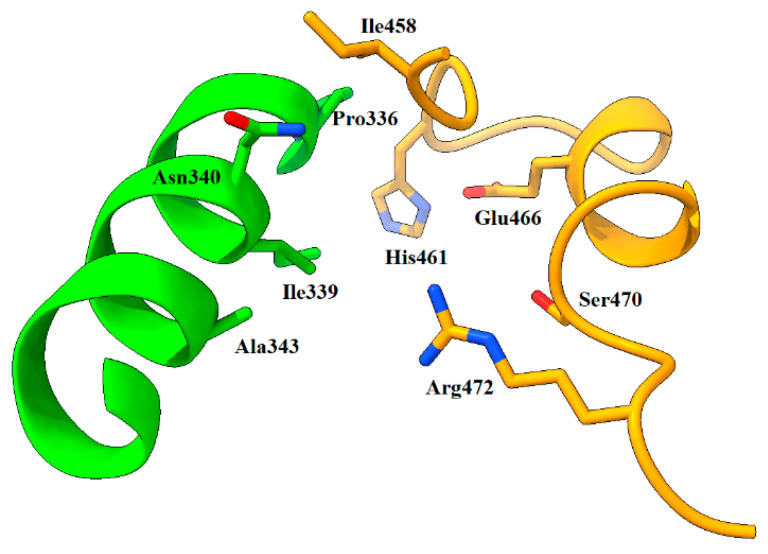
The λ-site and its composition. Note that this pocket is located at the top of the catalytic site. Chain A = green, chain B = orange. Atom color: N (blue), O (red), S (yellow).

**Figure 9 ijms-24-16046-f009:**
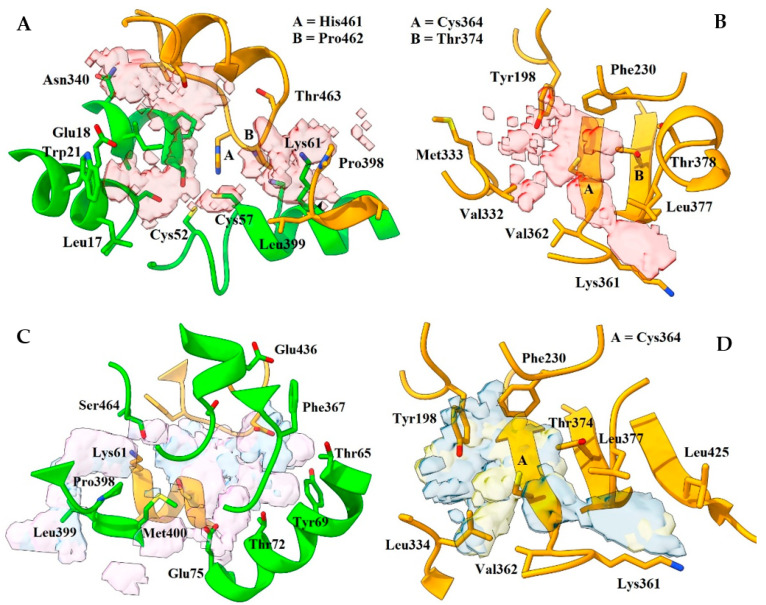
The top-ranked pockets found in TRLMS by cosolvent molecular dynamics. (**A**) 5 ns, cavity 00 (berry red); (**B**) 5 ns, cavity 02 (red); (**C**) 20 ns, cavity 00 (blue) and 40 ns, cavity 00 (purple); (**D**) 20 ns, cavity 02 (dark green) and 40 ns, cavity 03 (yellow green). Only the most representative residues are shown; more data regarding these pockets can be found in the Appendix A. Chain A = green, chain B = orange. Atom color: N (blue), O (red), S (yellow).

**Figure 10 ijms-24-16046-f010:**
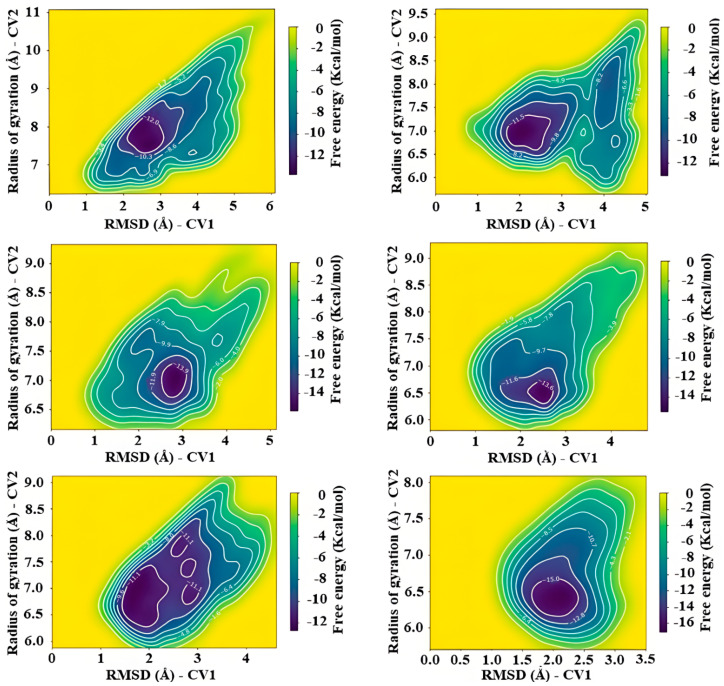
Free energy surfaces obtained from six independent runs of metadynamics simulations in TRLMS for the σ-site.

**Figure 11 ijms-24-16046-f011:**
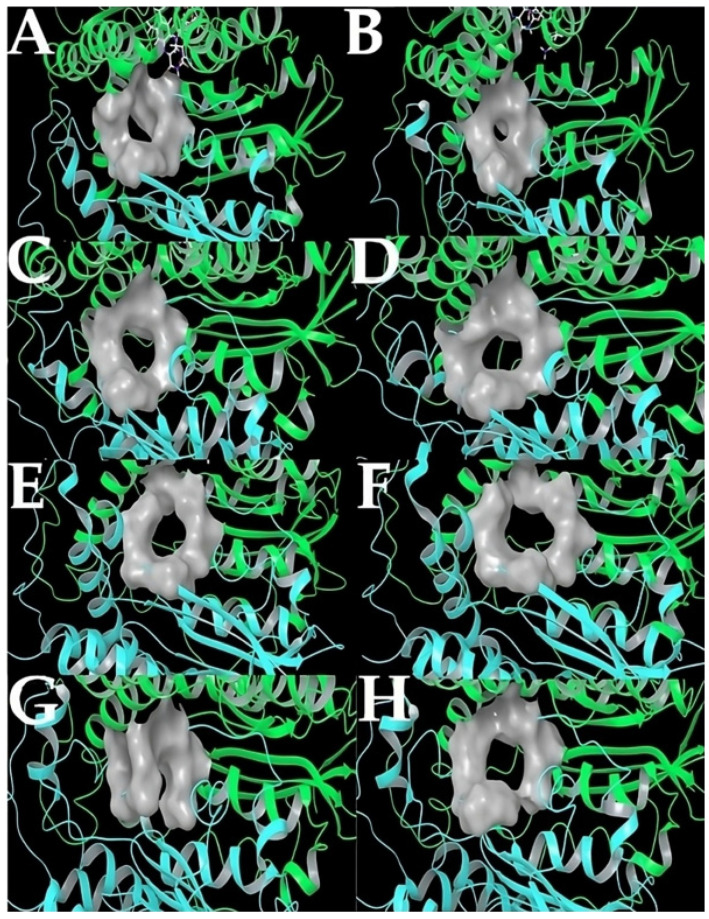
Surface representation of the residues belonging to the interface region (gray). Conformations belong to: region prior to metadynamics simulation (**A**,**B**), minimum values of CVs with a bias of 3.0 (**C**,**D**), 5.0 (**E**,**F**), and 8.0 (**G**,**H**), respectively. The structures of FAD and NADPH are seen as colored sticks. Chain A = green, chain B = cyan.

**Figure 12 ijms-24-16046-f012:**
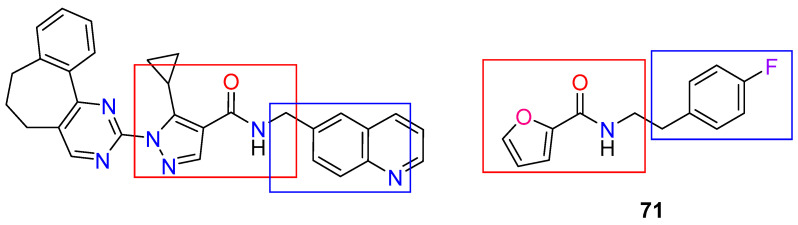
Structures and structural similarities between ZINC1215998 and compound **71**. The red and blue boxes indicate the fragments of ZINC12151998 that are similar in compound **71**.

**Figure 13 ijms-24-16046-f013:**
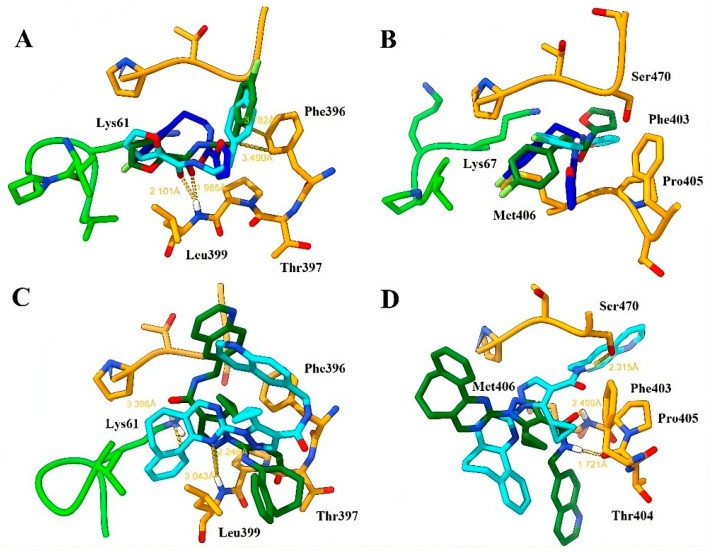
Poses obtained by LeDock and PLANTS in ZINC12151998 and compound **71** at the TRLMS and GR (PDBID code 3DJJ) σ-site. The native pose of **71** in PDBID code 5S9W is represented with blue sticks, while the poses obtained by LeDock and PLANTS with green and cyan sticks, respectively. Only the most representative residues and the distance in Å with the ligand are shown. (**A**,**C**) TRLMS; (**B**,**D**) GR (PDBID code 3DJJ). Chain A = green, chain B = orange. Atom color: N (blue), O (red), S (yellow) and F (light green).

**Figure 14 ijms-24-16046-f014:**
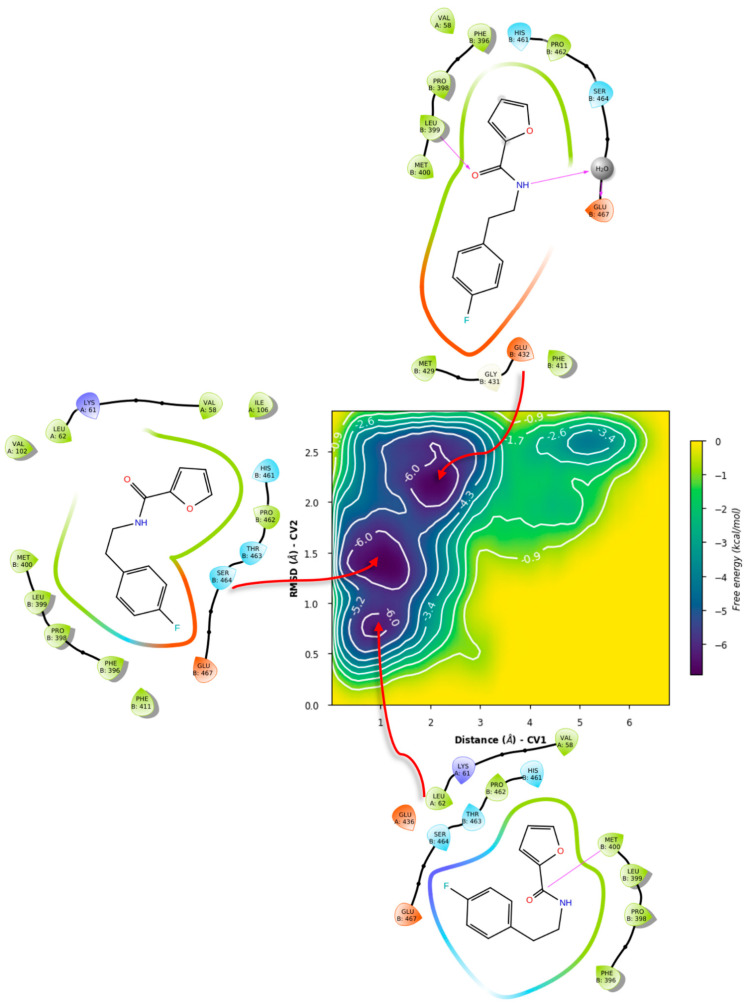
Free energy surface for PLANTS’ top pose of compound **71** in TRLMS.

**Figure 15 ijms-24-16046-f015:**
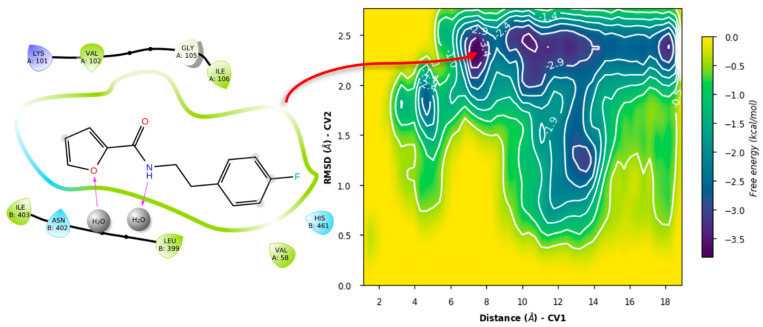
Free energy surface for LeDock’s top pose of compound **71** in TRLMS.

**Figure 16 ijms-24-16046-f016:**
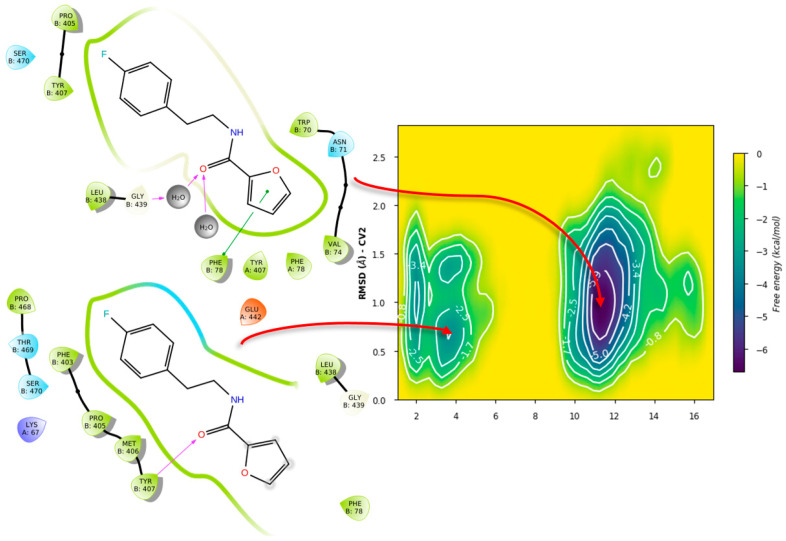
Free energy surface for PLANTS’ top pose of compound **71** in GR.

**Figure 17 ijms-24-16046-f017:**
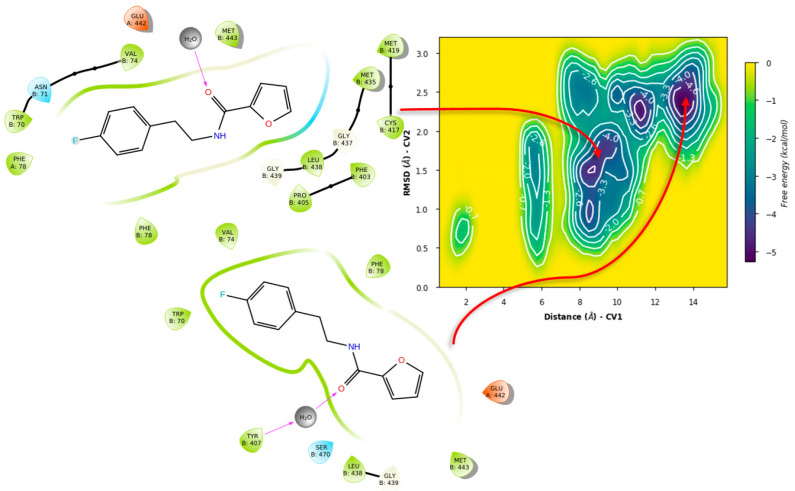
Free energy surface for LeDock’s top pose of compound **71** in GR.

**Figure 18 ijms-24-16046-f018:**
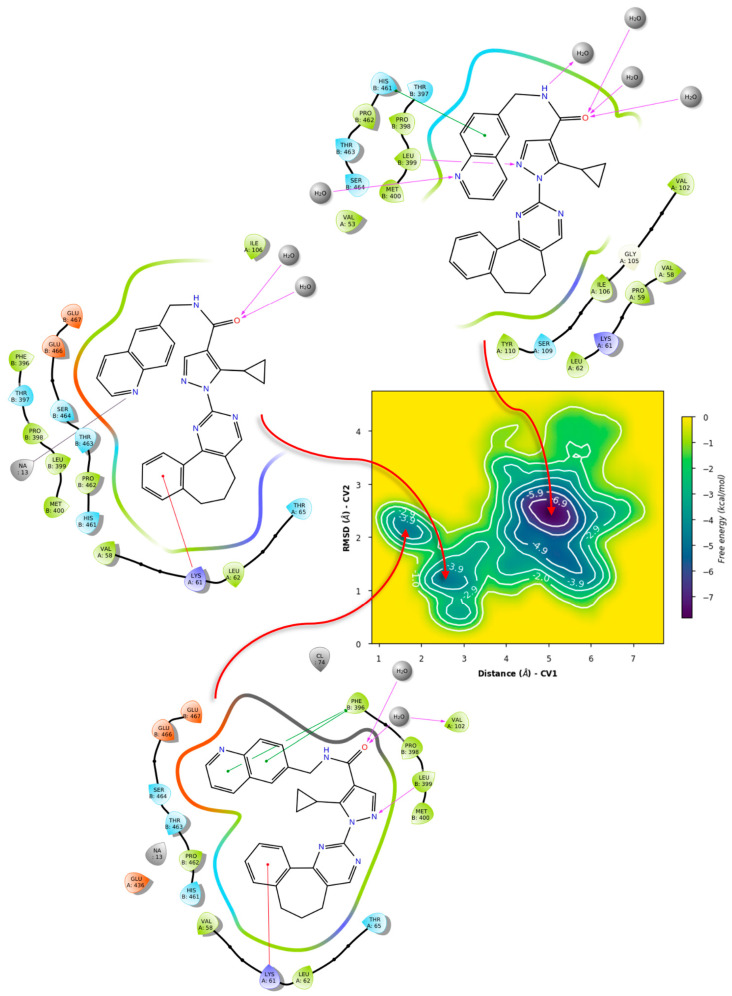
Free energy surface for PLANTS’ top pose of ZINC12151998 in TRLMS.

**Figure 19 ijms-24-16046-f019:**
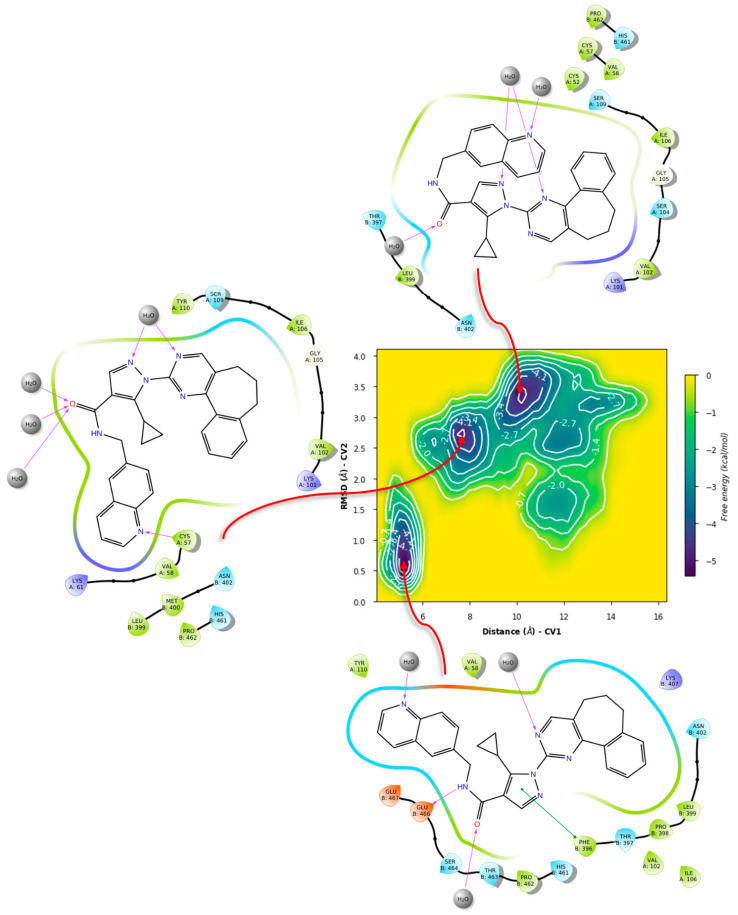
Free energy surface for LeDock’s top pose of ZINC12151998 in TRLMS.

**Figure 20 ijms-24-16046-f020:**
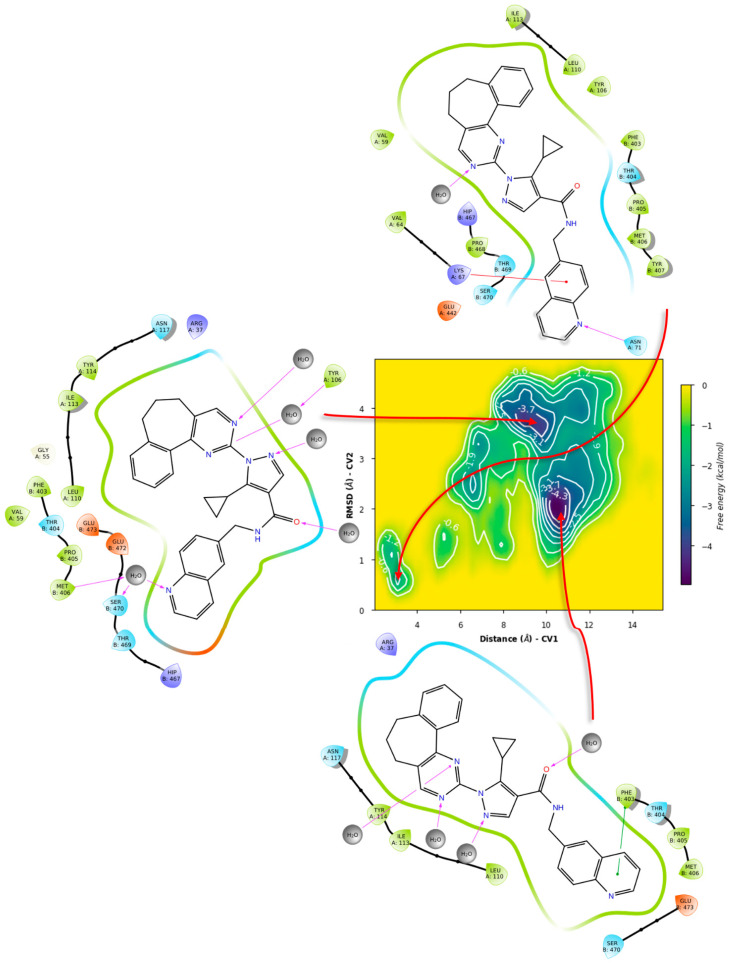
Free energy surface for PLANTS’ top pose of ZINC12151998 in GR.

**Figure 21 ijms-24-16046-f021:**
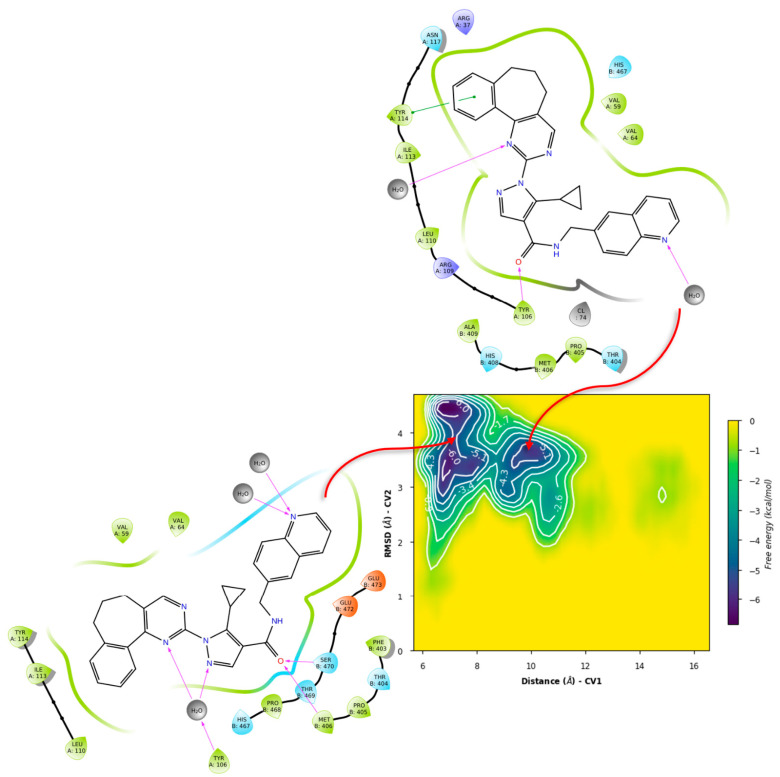
Free energy surface for LeDock’s top pose of ZINC12151998 in GR.

## Data Availability

The data are contained in this article and the Appendix A.

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
