# Peer review of "In Silico Exploration of the Trypanothione Reductase (TryR) of L. mexicana"

_ijms, 2023, doi:10.3390/ijms242216046_

Round 1

Reviewer 1 Report

Comments and Suggestions for Authors

The manuscript by Francisco et al offers a commendable deep dive into the role of the polyamine metabolic pathway in leishmaniasis, emphasizing the significance of trypanothione reductase. The authors rightly highlight the challenges of conventional molecular docking techniques and introduce innovative computational methods to uncover new potential drug targets, especially the promising σ-site. This work promises to be a valuable addition to the current body of literature on leishmaniasis, its molecular mechanisms, and drug development endeavors. The manuscript is well-written and organized. Thus, this reviewer recommends its publication on Int. J. Mol. Sci. without any changes. 

Author Response

Dear reviewer

I appreciate your positive comments on the text, our working group worked hard for this result.

Greetings.

Reviewer 2 Report

Comments and Suggestions for Authors

The authors extensively use tools and methods to find cryptic pockets.

The system is clearly presented in the introduction and the aim of the work is coherent with the methods used. I have just some minor observations regarding the application of MD/metadynamics simulations.

Details regarding MD simulations should be added. In particular, there is no indication of the force field used in the simulations and how the systems with the cosolvents were generated (software and parametrization of cosolvents).

Some references should be added when indicating MD integrator, thermostat and barostat used in simulations.

Figure S16: some of the FES present points in the CVspace, while other not: explain or remove the points?

The authors explain that they use metadynamics just to confirm an overall sampling of the cryptic pockets. However, they used highly degenerate CVs. Even without reaching or claiming convergence, the authors could add a cluster analysis of snapshots, pooled from simulations free-energy minima, to provide significant and alternative conformation of these residues and eventually comparing them with ones retrieved from unbiased simulations.

Author Response

Dear reviewer:

We appreciate the observations made to manuscript and in this response we list the corrections made to it in accordance with the comments made:

The software, force field and solvent parameterization used for cosolvent dynamics were specified more explicitly:

a) Software: Section 3.5., page 14, reference 106.
b) Force field: Section 3.2., page 13, reference 108.
c) Parameterization of solvents: Section 3.5., page 14, reference 122.

The references of the MD integrator, thermostat and barostat used in the simulations were also specified more clearly:

a) Integrator: Section 3.2., page 13, reference 117.
b) Thermostat: Section 3.2., page 13, reference 118.
c) Barostat: Section 3.2., page 13, reference 119.

The points in Figure S16 of the supplemental material were removed.

For the expansion in the metadynamics analysis, histograms were made that were included in support information (Figure S19 and S20) comparing the trends of the radius of gyration of the classical simulations and comparing it with the metadynamics results. Discussion of this is addressed in the last paragraph on page 12 of the manuscript.

I attach the document with all the changes mentioned above.

Once again, we appreciate your comments and are awaiting your responses.

Reviewer 3 Report

Comments and Suggestions for Authors

The authors describe a computational study focused on TryR from L. Mexicana aimed to identify promising druggable sites on this unexplored drug target. By integrating multiple structure-based analyses the authors identify three interesting hot-spots and select one of these, the sigma-site, as the most favorable for drug development.

While this is a well-executed computational analysis, I believe it is too preliminary to justify publication. In my opinion, besides the computational identification of druggable sites (that, indeed, are not very new because already reported for TryR from L. infantum and T. brucei, as stated by the authors) the study should include some kind of experimental validation. For example, the authors could perform a high-throughput virtual screening of commercial compounds focused on the selected sites followed by experimental testing for activity or binding.
Even if the study stays “all computational” the analysis should be expanded to make it significant and helpful to the scientific community. Considering the high degree of conservation of TryR (about 90% for Leishmania and >65% for Trypanosoma spp, and close to 100% for the substrates binding cavities), it would be important to check if the selected sites are conserved in the other trypanosomatidae spp, revealing if any of them can can be exploited for the development of broad-spectrum trypanocidal agents. Moreover, the selectivity of the pockets should be checked, i.e. the structural comparison should include the main off-target of TryR, namely human GR. In particular the region of the trypanothione binding cavity including the sigma-site is rather conserved in GR (lines 186-187: glutathione moiety binds there) thus the design of inhibitors targeting this site should be carefully evaluated. Indeed, a potent and selective inhibitor of TryR from L. infantum and T. brucei targeting the region including the sigma-site has been recently reported (
https://doi.org/10.1021/acsinfecdis.2c00325), the molecular bases for selectivity are discussed by comparison of crystal structures of TryR and GR (discussed here too doi: 10.3389/fmolb.2022.900882) but a computational analysis (MD) could be interesting.

Point-by-point remarks:

Line 83: percentage of residual activity without concentration of inhibitor is meaningless

Figure 2: the figure is poorly informative and its purpose is unclear in the context of this study.

Line 99 and elsewhere: interphase is a typo for “interface”, please correct

195-197: integrating analyses from different algorithms with complimentary approaches is good practice, no need for justification!

200: does the site correspond to the so-called “doorstop pocket” reported for TryR?

Figure 4: it should be improved. I suggest showing the overall as light-grey surface rather than cartoon, stick colored probes for FTMap. Zoomed views should be bigger, labels are difficult to read. Blue frame is unnecessary.

289-298: I am not sure I understood your choice. If you are looking for general hot-spots the probes should be chosen to cover the widest chemical space. If, instead, you are looking for “guidelines for optimization of the original lead” you should comment on that: the results are in line with your previous work?

Figure 9: please state in the caption that it refers to the sigma-site only.

Figure 10: it is difficult to appreciate the evolution of the pocket if any panel shows a different view. Please choose one orientation for all. Specify in the caption which panels refer to TRLMS of 2W0H. Panel C has different colors.

Supplementary session is confusing, the content of each paragraph should be better explained and reorganized to make it useful to the reader. Some (not exaustive) comments:

Table S3: non bonded interactions with what? If the table lists the residues shaping the pockets identified by FTMap it would be more informative to group the residues by pocket rather than list them by sequence order.

Table S5: according to FTMap  it seems that 2W0H has no druggable pockets but this is not the case. Could you comment on this?

Table S7: is there some correspondence with the pockets discussed in the main text (sigma, lambda, NADPH….)? please add this information to the table.

Comments on the Quality of English Language

The text is clear, few typos are present.

Author Response

Dear reviewer:

We appreciate the observations made to manuscript and in this response we list the corrections made to it in accordance with the comments made:

  1. In my opinion, besides the computational identification of druggable sites (that, indeed, are not very new because already reported for TryR from L. infantum and T. brucei, as stated by the authors) the study should include some kind of experimental validation. For example, the authors could perform a high-throughput virtual screening of commercial compounds focused on the selected sites followed by experimental testing for activity or binding.

We consider that, due to the origin and choice of ZINC12151998, which comes from a docking of 600,000 compounds, this part has been covered. The explanation is found in lines 305-307.

  1. Even if the study stays “all computational” the analysis should be expanded to make it significant and helpful to the scientific community. Considering the high degree of conservation of TryR (about 90% for Leishmania and >65% for Trypanosoma spp, and close to 100% for the substrates binding cavities), it would be important to check if the selected sites are conserved in the other trypanosomatidae spp, revealing if any of them can can be exploited for the development of broad-spectrum trypanocidal agents. 

The discussion of this point has been placed on lines 430-452. Additionally, Table S11 has been attached for this purpose.

  1. Moreover, the selectivity of the pockets should be checked, i.e. the structural comparison should include the main off-target of TryR, namely human GR. In particular the region of the trypanothione binding cavity including the sigma-site is rather conserved in GR (lines 186-187: glutathione moiety binds there) thus the design of inhibitors targeting this site should be carefully evaluated. Indeed, a potent and selective inhibitor of TryR from L. infantum and T. brucei targeting the region including the sigma-site has been recently reported (https://doi.org/10.1021/acsinfecdis.2c00325), the molecular bases for selectivity are discussed by comparison of crystal structures of TryR and GR (discussed here too doi: 10.3389/fmolb.2022.900882) but a computational analysis (MD) could be interesting.

The discussion of this point is located on lines 453-470. We have considered that there are sufficient reasons and evidence in the literature to believe that, although GR has many similarities with TryR, a ligand designed based on TryR should not exhibit promiscuous behavior with GR. Additionally, Table S12 has been attached for this purpose, compared both sequences.

  1. Line 83: percentage of residual activity without concentration of inhibitor is meaningless

The correction is found on line 82-83.

  1. Figure 2: the figure is poorly informative and its purpose is unclear in the context of this study.

The justification for the use of this figure is placed on lines 91-93.

  1. Line 99 and elsewhere: interphase is a typo for “interface”, please correct

Interphase was corrected in all text by interface.

  1. 195-197: integrating analyses from different algorithms with complimentary approaches is good practice, no need for justification!

The justification was removed from the text.

  1. 200: does the site correspond to the so-called “doorstop pocket” reported for TryR?

The explanation at this point is found in lines 204-207, where it is explained that the remains found are nearby but not necessarily part of the doorstop pocket.

  1. Figure 4: it should be improved. I suggest showing the overall as light-grey surface rather than cartoon, stick colored probes for FTMap. Zoomed views should be bigger, labels are difficult to read. Blue frame is unnecessary.

Based on this comment, it was decided to make a general correction to all the images in the article, both those of the main text and those of the supplementary information.

  1. 289-298: I am not sure I understood your choice. If you are looking for general hot-spots the probes should be chosen to cover the widest chemical space. If, instead, you are looking for “guidelines for optimization of the original lead” you should comment on that: the results are in line with your previous work?

This point was discussed in lines 368-379, where it is clarified that, indeed, the computational results are in harmony with the previous work carried out.

  1. Figure 9: please state in the caption that it refers to the sigma-site only.

In Figure 9 (now Figure 11) the clarification was made that said image referred to the sigma site.

  1. Figure 10: it is difficult to appreciate the evolution of the pocket if any panel shows a different view. Please choose one orientation for all. Specify in the caption which panels refer to TRLMS of 2W0H. Panel C has different colors.

Figure 10 (now Figure 12) was made again so that the results described in the text could be in line with the visual aspect.

  1. Supplementary session is confusing, the content of each paragraph should be better explained and reorganized to make it useful to the reader. Some (not exaustive) comments:

13a. Table S3: non bonded interactions with what? If the table lists the residues shaping the pockets identified by FTMap it would be more informative to group the residues by pocket rather than list them by sequence order.

In Table S3 the clarification was made that these non-bonding interactions referred to non-covalent interactions. Additionally, in this and all tables, important residues were regrouped by area or pocket for greater clarity.

13b. Table S5: according to FTMap  it seems that 2W0H has no druggable pockets but this is not the case. Could you comment on this?

The clarification of this point was made in lines 215-217, which is attributed to the limitations of FTMap to detect hot spots in a static structure.

13c. Table S7: is there some correspondence with the pockets discussed in the main text (sigma, lambda, NADPH….)? please add this information to the table.

Like Table S3, Table S7 and others were corrected by specifying in their content whether the waste they collected was part of any important pocket found or already existing.

In addition to all the corrections already mentioned, an exhaustive review of the supplementary material was also carried out, correcting figures, titles and information at the bottom of the table.

Once again, we appreciate your comments and are awaiting your responses.

Round 2

Reviewer 3 Report

Comments and Suggestions for Authors

The previously identified compound ZINC12151998 can not be considered an experimental validation of the current prediction. Anyway, as I already stated, the paper can stay all-computational, given that the authors expand the analysis.

I appreciate the addition of the paragraph 2.7 where the authors present a comparative analysis of the pockets with respect to TryR from other species and the off-target hGR. However, some points should be improved.

Table S11 and Table S12  are not present in the Supplementary information.

443-444: “This would make it difficult to create a selective, high affinity and broad spectrum ligand for this area.” This is not true: selectivity (in the sense of distinguishing parasite from host) and affinity have nothing to do with the degree of conservation of a pocket, only the possibility to develop a broad-spectrum inhibitor has to do with that. In general, your discussion insinuates that a conserved pocket is a better target for inhibitor development but it is not a matter of better or worse: a pocket can be interesting and useful even if it is suited to development of specific drugs rather than broad-spectrum ones! What I want to know as a reader is “this druggable pocket is totally conserved thus suited for broad spectrum” or “this druggable pocket is absent in other species so it is very specific to L Mexicana” or “this pocket has differences that do/don’t prevent broad spectrum”. Indeed, differences can be more or less significant: for example the nadph cavity seems to have a certain variability but the substrate binds anyway, suggesting that differences must be not so relevant.

Comparison to hGR: it has been known for a long time that the trypanothione binding cavity has significant differences with respect to the glutathione binding cavity of hGR. The authors should specifically discuss the newly identified subpocket sigma-site: the (likely tiny) differences regarding specifically this site consent to design/identify a selective small ligand? Authors say only that they are very similar but do not discuss how similar. As a reader I would like to know if I can develop a selective drug just targeting this specific subpocket or not and what I should pay attention to. Even if the answer is no, this doesn’t mean that the pocket should be discarded: for example, an unselective small ligand targeting the sigma-site could be merged with a selective ligand targeting another subpocket in the cataliic cavity resulting in a potent and selective drug-like compound. As I already suggested, it would be interesting to compare the sigma-site of TryR and GR by MD simulations.

Figure 2: I understand that you want to keep this figure, but it is still poorly meaningful to me such that you could even remove the figure without affecting the manuscript. If you want to keep it you should at least expand the caption to explain how you selected them (all crystal structures available for TryR inhibitors from all sources?), how they are grouped (binding site? Which bind where?).

Are the authors sure that the zone 198, 230, 364 and 374 (first zone) in 2W0H, does not correspond to the doorstop pocket?

Interfase à correct with “interface”

Fig. S7 no letters in the panel. I expect to find in the figure all the “important residues” mentioned in the main text, visibly shown as sticks.

Possibly the authors uploaded an old version of the Supplementary?

Comments on the Quality of English Language

minor revisions needed.

Author Response

Dear reviewer:

In view of the latest comments made by you to our work, we have decided to  respond to each of them by making them effective in our text. Each of the points requested is addressed below:

  1. The previously identified compound ZINC12151998 cannot be considered an experimental validation of the current prediction. Anyway, as I already stated, the paper can stay all computational, given that the authors expand the analysis.

Answer: The compound ZINC12151998 was not considered a replacement for experimental validation, but the background of its candidate selection was included as a way to show that, at least virtually, the target cavity was closest to the trypanothione reductase (TryR) homologue of L. mexicana, that is, the TryR of L. infantum, had been explored (lines 296-298). In any case, the request for an expanded analysis of the target was made.

  1. I appreciate the addition of the paragraph 2.7 where the authors present a comparative analysis of the pockets with respect to TryR from other species and the off-target hGR. However, some points should be improved. 

a) Table S11 and Table S12 are not present in the Supplementary Information.

Answer: This information was reviewed and it was found that a version of the previous Support Information was uploaded. This has already been corrected by uploading the most corrected and current version of the Support Information.

b) 443-444: “This would make it difficult to create a selective, high affinity and broad spectrum ligand for this area.” This is not true: selectivity (in the sense of distinguishing parasite from host) and affinity have nothing to do with the degree of conservation of a pocket, only the possibility to develop a broad-spectrum inhibitor has to do with that. In general, your discussion insinuates that a conserved pocket is a better target for inhibitor development but it is not a matter of better or worse: a pocket can be interesting and useful even if it is suited to development of specific drugs rather than broad-spectrum ones! What I want to know as a reader is “this druggable pocket is totally conserved thus suited for broad spectrum” or “this druggable pocket is absent in other species so it is very specific to mexicana” or “this pocket has differences that do/don’t prevent broad spectrum”. Indeed, differences can be more or less significant: for example, the NADPH cavity seems to have a certain variability but the substrate binds anyway, suggesting that differences must be not so relevant.

Answer: In lines 419-442 this question is answered with the comparison of the residues in the critical areas of the target (TRLMS) with the TryR of other species (L. infantum, T. cruzi and T. brucei), demonstrating that the σ-site it is conserved in all parasites and strengthening the preliminary premise of being able to develop a broad-spectrum drug based on this. Table S11 was included in the Supporting Information to contemplate the aforementioned comparison.

c) Comparison to hGR: it has been known for a long time that the trypanothione binding cavity has significant differences with respect to the glutathione binding cavity of hGR. The authors should specifically discuss the newly identified subpocket sigma-site: the (likely tiny) differences regarding specifically this site consent to design/identify a selective small ligand? Authors say only that they are very similar but do not discuss how similar. As a reader I would like to know if I can develop a selective drug just targeting this specific subpocket or not and what I should pay attention to. Even if the answer is no, this doesn’t mean that the pocket should be discarded: for example, an unselective small ligand targeting the sigma-site could be merged with a selective ligand targeting another subpocket in the catalytic cavity resulting in a potent and selective drug-like compound. As I already suggested, it would be interesting to compare the sigma-site of TryR and GR by MD simulations.

Answer: The discussion of the comparison between trypanothione reductase (TryR) and glutathione reductase (GR) was divided into several subsections of section 2.7. The first of them consisted of the global comparison of both targets in critical characteristics related to the amino acid sequence (Supporting Information, Table S12) and the differences in the properties acquired by each target as a result of this (lines 443-461); additionally, sources were cited where the inhibitors designed for TryR had little or no effect on GR (lines 453-456). The second section consisted of the introduction of a metadynamics analysis of the GR σ-site and its comparison with that carried out on the TryR of L. mexicana (lines 462-466). Finally, in the third section, the behavior of the σ-site in TryR and GR in the presence of representative ligands was examined and investigated; to do this, they were first docked at the site of interest (lines 467-504) and then subjected to metadynamics with its corresponding discussion (lines 505-585). All of the above led to finding significant differences in the behavior of both cavities, which were expressed in Figures 14-21 of the main document and in the addition of extra material in the Supporting Information (Figures S22-S32).

  1. Figure 2: I understand that you want to keep this figure, but it is still poorly meaningful to me such that you could even remove the figure without affecting the manuscript. If you want to keep it you should at least expand the caption to explain how you selected them (all crystal structures available for TryR inhibitors from all sources?), how they are grouped (binding site? Which bind where?).

Answer: Figure 2 was removed from the text.

  1. Are the authors sure that the zone 198, 230, 364 and 374 (first zone) in 2W0H, does not correspond to the doorstop pocket?

Answer: The more in-depth review of the article by Fiorillo and collaborators (Front. Mol. Biosci. 2022, 9, 900882) allowed us to find that, indeed, residues 198, 230, 364 and 374 do belong to the doorstop pocket (lines 198-200).

  1. Interfase a correct with “interface”

Answer: All words written as "interfase" were corrected to "interface".

  1. S7 no letters in the panel. I expect to find in the figure all the “important residues” mentioned in the main text, visibly shown as sticks. Possibly the authors uploaded an old version of the Supplementary?

Answer: This has already been corrected by uploading the most corrected and current version of the Support Information.

We hope that the corrections made and detailed here cover the necessary requirements. Thank you for your attention. 

Round 3

Reviewer 3 Report

Comments and Suggestions for Authors

I thank the authors for expanding the analysis according to my suggestions. I believe that the manuscript has been substantially improved and it can be published in the present form.

Comments on the Quality of English Language

Good, apart from a few typos that should be corrected (afinity, basedon, ...).